# Effect of Fungal Endophyte *Epichloë bromicola* Infection on Cd Tolerance in Wild Barley (*Hordeum brevisubulatum*)

**DOI:** 10.3390/jof8040366

**Published:** 2022-04-02

**Authors:** Yurun Zhai, Zhenjiang Chen, Kamran Malik, Xuekai Wei, Chunjie Li

**Affiliations:** State Key Laboratory of Grassland Agro-Ecosystems, Key Laboratory of Grassland Livestock Industry Innovation, Ministry of Agriculture and Rural Affairs, Engineering Research Center of Grassland Industry (Ministry of Education), Gansu Tech Innovation Centre of Western China Grassland Industry, Center for Grassland Microbiome, College of Pastoral Agriculture Science and Technology, Lanzhou University, Lanzhou 730000, China; zhaiyr20@lzu.edu.cn (Y.Z.); chenzhj17@lzu.edu.cn (Z.C.); malik@lzu.edu.cn (K.M.); weixk17@lzu.edu.cn (X.W.)

**Keywords:** wild barley, grass-endophyte fungi symbiosis, heavy metal, Cd distribution, subcellular distribution, chemical forms analysis

## Abstract

Hydroponic *Hordeum brevisubulatum* (wild barley) was used as material in the greenhouse to study the effects of endophyte infection on plant growth, Cd absorption and transport, subcellular distribution, and Cd chemical forms under CdCl_2_ stress. Endophytic fungi respond positively to chlorophyll content and photosynthetic efficiency under Cd stress. The order of Cd absorption in different parts of the plant was: roots > stems > leaves. Endophyte infection increased the plant’s absorption and transport of Cd while causing a significant difference in the stem, which was associated with the distribution density of endophyte hyphae. The proportion of organelle Cd in endophyte-infected wild barley was significantly higher, which facilitated more Cd transport to aboveground. Cd stress showed a slight effect on the chemical forms of Cd in leaves. The proportion of phosphate, oxalate, and residual Cd increased in the stem. Cd existed in the form of inorganic salt, organic acid, pectin, and protein in roots. Endophyte infection reduced the Cd content of the more toxic chemical forms to protect the normal progress of plant physiological functions. Therefore, the isolation of cell walls and vacuoles is a key mechanism for plant Cd tolerance and detoxification. As endophyte infections have more ability to absorb Cd in plants, *H. brevisubulatum*–*Epichloë* *bromicola* symbionts can improve heavy metal contaminated soil and water.

## 1. Introduction

In recent years, due to the continuous increase of agricultural and industrial activities, a large number of heavy metals have been released into the atmosphere, water, and soil [1]. Cd is a toxic heavy metal and has become one of the four metals of global concern regarding the ecological environment and human health [2]. Cd is a non-essential nutrient element in plant growth and development, but it is easily absorbed by plants and produces toxic effects [3], such as reducing various photosynthetic parameters, inhibiting root growth, reducing dry and fresh weight, and decreasing genomic template stability [4,5]. Meanwhile, Cd may also affect humans through the food chain. Therefore, the remediation of heavy metal contaminated soil is key to sustainable agricultural development. There are various ways to implement remediation, such as chemical remediation technology, physicochemical remediation technology, and phytoremediation technology. However, the first two approaches are constrained by high costs, large-scale difficulties, and secondary pollution risks, and phytoremediation technology has the advantages of economy, efficiency, and environmental protection [6].

Symplastic and apoplastic refer to the ways for plant roots to absorb Cd, which are distributed aboveground through various transport pathways [7]. Simultaneously, plants initiate a series of strategies to tolerate Cd stress. Plants improve the toxicity and migration of Cd by regulating the chemical form of Cd [8]. The cell wall is the first structure that prevents heavy metals from entering the cell and its adsorption and chelation of heavy metals is a tolerance strategy to reduce heavy metal damage [9,10]. Cell walls are mainly composed of pectin, cellulose, hemicellulose, lignin, protein, and some functional groups. These functional groups form a large number of heavy metal binding sites and adsorb Cd on the cell wall, to hinder its transmembrane transport and reduce the damage caused by heavy metals to plants [8,11]. In addition, heavy metal stress can significantly increase the endocytic internalization of pectin, to promote the deposition and sequestration of Cd in the cell wall [12]. However, the metal ion binding sites on the cell wall are limited. When the tolerance limit is exceeded, the Ca^2+^, Fe^2+^, and Zn^2+^ transporters are used by Cd^2+^ to enter the cell, which is the molecular basis for plant cells to absorb Cd^2+^ [13,14]. The soluble fraction includes cytoplasm and cellsap. Vacuoles account for 90% of cell volume. After heavy metals enter the cell, heavy metal complexes containing sulfide can be formed in vacuoles to reduce toxicity. Therefore, vacuoles represent the second barrier for plant cells to resist heavy metals [14]. In addition, studies have shown that exogenous glutamate betaine, glutathione, secondary metals (Cu, Zn, Ca), EDTA, or elevated CO_2_ levels can effectively alleviate Cd-induced growth inhibition [15,16,17,18,19]. The use of endosymbiotic microbes to alleviate the damage caused by Cd is also an effective method. The inoculation of endophytic bacteria, such as SaMR12, IU01, and IU02, can produce phytohormone, phosphate solubilization, or regulate the glutathione-ascorbic acid cycle to stimulate an antioxidant response which helps in the improvement of plant growth and Cd resistance [20,21].

According to the principle of “one fungus, one name”, the asexual (anamorphic) taxa and the sexual (teleomorphic) taxa are collectively referred to as *Epichloë* [22]. Endophytic fungi colonize the intercellular space, and hyphae are distributed in all above-ground parts of the plant [23]. Endophytic fungi spread vertically, from seed to offspring through the mother plant, and horizontally, which can colonize the plant above ground asymptomatically. Of course, vertical transmission is the main mode of endophytic fungal transmission [24,25]. *Epichloë* endophytes often form symbionts with cool-season grasses (subfamily Poöideae) [26]. The grasses do not show any symptoms [24,27] but are endowed with a variety of resistance by *Epichloë* endophytes, including pathogens, insects, waterlogging, drought, cold, salt, and heavy metals, etc. [28,29], which are the advantages of *Epichloë* endophytes that produce alkaloids or secondary metabolites to the host [30]. The remediation effect of plants on heavy metal contaminated soil or water mainly depends on the absorption and accumulation of heavy metals by plants. The number of tillers and biomass of meadow fescue (*Festuca pratensis*) and tall fescue (*Festuca arundinacea*) infected by *Epichloë* endophytes (E+) can significantly increase under Cd stress (*p* < 0.05) in comparison with endophyte-free (E−. The accumulation of Cd increases in roots and buds, which improves the transport of cadmium from roots to stems, and shows a higher ability to remove Cd from soil and aqueous solutions [31,32]. Endophyte infection is considered beneficial for the growth and anti-oxidant mechanism of the host exposed to high concentrations of CdCl_2_. The germination rate, germination index, plant height, root length, biomass, and tiller number of endophyte-infected drunken horse grass (*Achnatherum inebrians*) (100, 200 µM) and *Elymus dahuricus* (100, 200, 300 µM) significantly increase under high concentrations of Cd stress. In addition, the antioxidant enzyme activity (AEA), proline content, malondialdehyde (MDA) content, H_2_O_2_ concentration, and chlorophyll level also increase significantly (*p* < 0.05) [33,34]. Meanwhile, *Epichloë* endophytes also enhanced the zinc tolerance of perennial ryegrass (*Lolium perenne*) and aluminum tolerance of fescues (*Festuca* spp.) [35,36]. Therefore, grass and *Epichloë* endophyte symbionts have great potential for the remediation of heavy metal contaminated soil and water.

Wild barley (*Hordeum brevisublatum*) is a universal high-quality forage that is widely distributed in the lightly salinized meadows of northern China. It is well known for its high resistance to abiotic stress, including drought, salt, and alkali tolerance [37]. Previous studies have demonstrated that the characteristics and functions of *Epichloë* endophytes in wild barley make *Hordeum brevisubulatum*–*Epichloë bromicola* symbionts more competitive, and the salt tolerance is the most prominent [38,39,40], which indicates that the *H. brevisubulatum*–*E. bromicola* symbionts demonstrate a certain absorption of metal (Na). Therefore, as an ecological grass, the wild barley endophyte symbionts play a vital role in improving the environment and repairing contaminated soil. However, the adsorption capacity of *H. brevisubulatum*–*E. bromicola* symbionts to heavy metals needs to be further explored.

In the current study, a hydroponic growth system was adopted to reduce the interaction between soil substrate and Cd and to better observe the morphological characteristics of plants after absorbing Cd. The main objective was to explore whether *Epichloë* endophyte infection affects the growth and development of host *H. brevisubulatum* treated with various concentrations of CdCl_2_ and to clarify the distribution of Cd in the host. Thus, the possible mechanism of endophytic fungi affecting the distribution of host Cd was explained, which laid a foundation for wild barley endophyte symbionts as a remediation plant to improve the soil polluted by heavy metals.

## 2. Materials and Methods

### 2.1. Plant Materials and Growth Conditions

The seeds of *H. brevisubulatum* were harvested in Linze County, Gansu Province, China. The sequences were submitted in Genbank with accession numbers KU365146-148 (*tefA*) and KU365152-154 (*actG*). A phylogenetic tree reference was adopted from the work of Chen [41]. The seed samples were stored at 4 °C in a seed bank to maintain the endophyte viability. In the greenhouse, both E+ or E− seeds were respectively sown in two-hole trays containing sterilized vermiculite and watered as needed. The seeds were cultured for 2 weeks after germination, and the seedlings with similar growth vigor were rinsed and transplanted into opaque hydroponics pots for water culture. Each hydroponic pot contained 5 holes with a capacity of 3 L and an aerating pump inside. Wild barley seedlings were grown under greenhouse conditions with the same volume of half-strength Hoagland nutrient solution. To keep the balance of ion nutrition, the nutrient solution was renewed every 7 d. After 4 weeks, there were 3 groups of E+ and E− hydroponic pots, respectively, for a total of six groups, each in quadruplication. There were two plastic pots in a repeat and 58 pots in total. CdCl_2_ was gradually applied in half Hoagland’s nutrient solution, and the concentration gradients were set as 0, 50, 100 µM. The treatment solution was changed at every 3 d to maintain the precise Cd concentrations in the solution.

After 6 weeks of exposure to CdCl_2_ stress, the plant roots were soaked in 20 mM ethylenediamine tetraacetic acid disodium salt (EDTA-2Na) for 15 min to remove ions from the root surface and then repeatedly washed with distilled water for 3 times. The roots, stems, and leaves of the plant were harvested separately.

### 2.2. Experimental Design

#### 2.2.1. Plants Growth and Photosynthetic Parameters

Plant photosynthetic indexes were measured before harvesting using a GFS-3000 (Walz, Bayern, Germany) portable photosynthesis-fluorescence measurement system. The SPAD-502 Plus (Konica Minolta, Tokyo, Japan) chlorophyll meter was used to determine the chlorophyll content of plants. At the same time, the plant height, root length, and tiller number were measured and the fresh weight of roots, stems, and leaves was recorded. All tissues were dried to a constant weight at 65 °C and used for the determination of Cd content after grinding into powder.

#### 2.2.2. Cd Distribution in Different Parts

Chen’s [42] method was slightly modified to determine Cd concentrations in plants. Briefly, a 0.5 g plant sample was weighed and added into a 100 mL digestion tube. A 10 mL digestion agent (nitric acid: concentrated sulfuric acid: perchloric acid = 8:1:1, *V:V:V*) was added to the solution and placed on the graphite digestion instrument for digestion at 370 °C. After complete digestion and cooling, the tube wall was repeatedly washed and the volume adjusted in a 50 mL volumetric flask. The flame atomic absorption spectrometer (Thermo, Waltham, MA, USA) was used to determine the Cd content in the solution.

#### 2.2.3. Subcellular Distribution of Cd in *H.*
*brevisubulatum*

The subcellular extraction and separation in plant tissues were carried out by Weigle and Jager’s differential centrifugation method [43]. Briefly, plant tissues (0.25 g) were homogenized by adding 15 mL of pre-cooling extraction buffer (50 mM Tris-HCl buffer solution (pH = 7.5), 250 mM sucrose, 1 mM Dithioerythritol). The homogenate was centrifuged at 3000 rpm for 15 min, and the precipitation was cell wall component (FI). Then, the supernatant was centrifuged for half an hour (at 15,000 rpm). The precipitates were referred to as the organelle component (FII) and the resulting supernatant was the soluble fraction (including high molecular and macromolecular organic substances and inorganic ions in cytoplasm and vacuole, FIII). All steps were performed at 4 °C. The FI, FII, and FIII components were oven-dried at 60 °C to determine the Cd content.

#### 2.2.4. Chemical Forms Extraction

Wu’s method was adopted to determine the chemical forms of Cd in different parts of E+ and E− plants [44]. The experimental procedures were carried out in sequence according to the specified solution:80% ethanol, extracting inorganic Cd, giving priority to nitrate/nitrite, chloride, and aminophenol cadmium, i.e., 80% ethanol extraction state (F_Ethanol_).Deionized water, extracting water-soluble Cd, Cd-organic acid complexes, and Cd(H_2_PO_4_)_2_, i.e., deionized water extraction state (F_H2O_).1 M NaCl, extracting pectates and protein-integrated Cd, i.e., 1 M NaCl extraction state (F_NaCl_).2% acetic acid, extracting undissolved cadmium phosphate, including CdHPO_4_, Cd_3_(PO_4_)_2_, and other Cd-phosphate complexes, i.e., 2% HAc extraction state (F_HAc_).0.6 M HCl, extracting cadmium oxalate, i.e., 0.6 M HCl extraction state (F_HCl_)The remainder is the residual state (F_R_).

The fresh plant tissue was ground in the extract to obtain a homogenate, shaken well at 25 °C for 24 h, and the supernatant was carefully collected. The extractant was added again, shaken for two hours, and the supernatant collected. This process was repeated twice, and the collected supernatants were mixed. The residue was added to the next extractant and the fore-mentioned steps were repeated. Subsequently, the chemical reagents were extracted step by step, dried at 70 °C, and digested according to Section 2.2.2.

### 2.3. Statistical Analyses

SPSS statistical software (Ver. 19.0, SPSS, Inc., Chicago, IL, USA) was used for the analysis of variance (ANOVA) and Duncan’s multi range test. Two-way analysis of variance was applied to determine the effects of *Epichloë* endophyte (E) and Cd stress (S) on plant growth and development, photosynthesis, Cd distribution in different parts of plants, subcellular distribution, and Cd chemical forms. Statistical significance was defined at the 95% confidence level. All measurements were shown as mean ± standard errors.

## 3. Results

### 3.1. Plants Growth and Photosynthetic Parameters

Plant growth and photosynthesis were inhibited under Cd treatment. Compared with the control, Cd stress significantly (*p* < 0.05) reduced the plant height, root length, dry weight, and fresh weight, tiller number (Table 1, Figure 1 and Figure 2A–E).

The number of tillers in E− plants was lower than E−, but there was no significant (*p* > 0.05) difference between them (Figure 2E). The plant height of E− plants was significantly (*p* < 0.05) higher than E+ under control condition (Figure 2A), but the presence of endophytic fungi had no significant (*p* > 0.05) effect on plant growth under other CdCl_2_ concentration gradients (Figure 2A–E). The content of chlorophyll was significantly (*p* < 0.05) reduced with increased CdCl_2_ stress, and no significant (*p* > 0.05) difference was observed between E+ and E− under the treatment of 0 and 50 µM CdCl_2_ concentrations. The chlorophyll content of E+ plants under 100 µM CdCl_2_ stress was significantly (*p* < 0.05) higher than E− plants at the same concentration (Table 1, Figure 2F).

CdCl_2_ stress significantly (*p* < 0.05) reduced the transpiration rate (E), stomatal conductance (Gs), and net photosynthetic rate (Pn) of the plants and increased the concentration of intercellular CO_2_ (Ci) (Table 2, Figure 3). Among the three indicators of E, Gs, and Pn, E+ plants were significantly (*p* < 0.05) higher than E− (Figure 3A–C), whereas for the Ci, E+ plants were significantly (*p* < 0.05) lower than E−, and no significant (*p* > 0.05) difference was observed between them with the aggravation of stress (Figure 3D). In two-way ANOVA, the interaction of endophytic fungi and Cd stress had significant effects on the plant height and net photosynthetic rate of wild barley (*p* < 0.05), but had no significant effects on other growth and photosynthetic indexes (Table 2).

### 3.2. Cd Distribution in Different H. brevisublatum Parts

The distribution of Cd in the plant and the transport between “roots-stems” and “stems-leaves” were affected by CdCl_2_ treatment (Figure 4, Table 3 and Table 4). Regardless of the treatment, the overall situation of Cd distribution in the plant was as follows: roots > stems > leaves, and exhibited significant (*p* < 0.05) differences (Figure 4, Table 3).

The Cd concentration in leaves, stems, roots, and the whole plants increased significantly (*p* < 0.05) with the increase of CdCl_2_ concentration, while the transport index (TI_roots-stems_, TI_stems-leaves_) was to the contrary. However, the above indicators all showed that E+ plants were higher than E−, but only with significant (*p* < 0.05) differences in the Cd concentration of stems (Table 3). In the control, more than 60% of the Cd was distributed in the roots, which was more prominent after treatment with CdCl_2_, reaching about 80%, significantly higher than the control (*p* < 0.05), and E+ was lower than E− but the difference was not significant (*p* > 0.05). The results for stems and leaves were to the contrary, as the increase in the CdCl_2_ concentration resulted in a significant (*p* < 0.05) decrease in the distribution proportion of Cd compared with the control. Moreover, E+ plants were higher than E−, but did not show significant (*p* > 0.05) differences (Figure 4). In two-way ANOVA, the interaction of endophytic fungi and Cd stress had no significant (*p* > 0.05) effects on Cd concentration in different parts of plants and transport index (TI_roots-stems_, TI_stems-leaves_) (Table 4).

### 3.3. Cd Subcellular Distributions and Proportions in H. brevisubulatum

The subcellular distribution of Cd concentration in the roots, stems, and leaves was different because of the different Cd treatments. In leaves, 40% or so of the Cd fraction in the cellular distribution was the soluble fraction, about 30% was the organelle fraction, and approximately 20% was the cell wall fraction (Figure 5). CdCl_2_ stress did not significantly (*p* > 0.05) affect the distribution proportion of Cd in the cell wall fraction and soluble fraction (Figure 5). However, the distribution proportion of Cd in the organelle fraction was significantly (*p* < 0.05) lower in E+ plants compared with the control under Cd stress, while E− plants had no significant (*p* > 0.05) difference (Figure 5). Under the condition of 100 µM CdCl_2_, the distribution proportion of Cd in organelle components in E+ plants was significantly (*p* < 0.05) lower than that in E− (Figure 5). CdCl_2_ treatment significantly (*p* < 0.05) increased the Cd concentration in the cell wall and soluble fraction, but had no significant (*p* > 0.05) effect on the organelle fraction (Figure 6A, Table 5). Endophyte had no significant (*p* > 0.05) effect on the subcellular distribution of Cd in leaves (Figure 6A).

The subcellular distribution proportion of Cd in plant stems was similar to that in leaves, and no significant (*p* > 0.05) effect of CdCl_2_ stress was detected on the distribution proportion (Figure 5). The distribution of Cd in cell walls, organelles, and soluble fractions was relatively uniform, each accounting for about 30% (Figure 5). The concentration of subcellular components in stems was less affected by Cd stress. In the organelle fraction, the Cd concentration of E− plants under the 50 µM CdCl_2_ condition was significantly (*p* < 0.05) lower than that of the control, and that of E+ plants under the 100 µM CdCl_2_ condition was significantly (*p* < 0.05) higher than that of the control (Figure 6B). In the soluble fraction, compared with the control under the condition of 100 µM CdCl_2_, the Cd concentration was significantly (*p* < 0.05) increased (Figure 6B). The overall performance of E− was higher than E+, and there were statistically significant (*p* < 0.05) differences in the 100 µM CdCl_2_ cell wall fraction, 0 µM CdCl_2_ organelle fraction, and 50 µM CdCl_2_ soluble fraction (Figure 6B). However, the interaction between endophyte and CdCl_2_ stress was not significantly different (*p* > 0.05) in two-way ANOVA (Table 5).

The subcellular distribution of Cd in roots was significantly affected by Cd stress (Figure 5 and Figure 6C, Table 5). The subcellular distribution of Cd under control conditions was similar to that in leaves. With increasing CdCl_2_ stress, the proportion of Cd distribution in the soluble fraction increased (from 40 to 70%) significantly (*p* < 0.05) and E+ plants were lower than E−, showing a significant (*p* < 0.05) difference at 50 µM CdCl_2_. However, the organelle fraction was significantly (*p* < 0.05) decreased (from 30% to 10%) and E+ plants were significantly (*p* < 0.05) higher than E− (Figure 5). The Cd concentration in the cell wall fractions of E− plants under 100 µM CdCl_2_ treatment was significantly (*p* < 0.05) increased (Figure 6C). The increase of CdCl_2_ concentration did not significantly (*p* > 0.05) affect the concentration of Cd in organelle fraction, but significantly (*p* < 0.05) increased the concentration of Cd in the soluble fraction (Figure 6C). The Cd concentration of the soluble fraction of E− plants was significantly (*p* < 0.05) lower than that in E+ plants under 0 µM CdCl_2_ conditions, but it increased significantly (*p* < 0.05) as Cd stress worsened in E− plants and was significantly higher than that of E+ plants (Figure 6C).

In two-way ANOVA, the interaction of *Epichloë* endophyte and Cd stress had no significant (*p* > 0.05) effects on the subcellular distribution of wild barley roots, stems, and leaves (Table 5).

### 3.4. Cd Chemical Forms and Proportions in H. brevisubulatum

In general, F_Ethanol_ and F_H2O_ forms of Cd were dominant in leaves and roots of *H. brevisubulatum*, and all forms were evenly distributed in the stems (Figure 7). The response of roots to Cd stress and endophytic fungi was the largest (Figure 7, Table 6).

In the leaves, approximately 85% of Cd forms were F_Ethanol_ and F_H2O._ The proportion of different chemical forms of Cd did not change significantly (*p* > 0.05) with increasing CdCl_2_ concentration (Figure 7, Table 7). However, the Cd concentration of F_Ethanol_ (CdCl_2_ at 50 and 100 µM) and F_NaCl_ (CdCl_2_ at 50 µM) forms increased significantly (*p* < 0.05) in E+ plants (Table 6). Endophyte had no significant (*p* > 0.05) effect on the concentration and percentage of different chemical forms of Cd in leaves (Table 6 and Table 7).

In the stems, the concentration and distribution proportion of different chemical forms of plants under CdCl_2_ stress are different under certain conditions (Figure 7, Table 6). F_NaCl_ (from 5.0% to 18.1%), F_HAC_ (from 2.7% to 15.5%), F_HCl_ (from 2.3% to 17.5%), and F_R_ (from 3.8% to 18.8%) forms of Cd were significantly (*p*
*<* 0.05) higher than in leaves and roots (Figure 7). Meanwhile, F_Ethanol_ and F_H2O_ decreased more significantly (*p*
*<* 0.05) than in leaves and roots, from 43.4% to 15.9% and from 42.8% to 13.6%, respectively (Figure 7). The proportion of F_HCl_ forms Cd under 100 µM CdCl_2_ stress showed that E+ was significantly (*p*
*<* 0.05) higher than E− (Figure 7). The concentration of F_NaCl_ forms Cd increased with the aggravation of stress, and E+ plants were significantly (*p*
*<* 0.05) lower than E− plants (Table 6).

In the roots, under the control condition, the distribution of Cd in each chemical form is similar to that in the stems, and the distribution proportion is relatively uniform (Figure 7). The proportion of F_Ethanol_, F_H2O_, and F_NaCl_ forms of Cd increased from about 60% to more than 80% with increasing CdCl_2_ concentration. After CdCl_2_ treatment, the proportion of F_HAC_, F_HCl_, and F_R_ forms Cd decreased significantly (*p* < 0.05) compared with the control (Figure 7). Under the control condition, the proportion of F_H2O_ and F_NaCl_ forms of Cd in E+ plants was significantly higher than that in E− (*p*
*<* 0.05), while the proportion of F_HCl_ and F_R_ forms of Cd was significantly lower than that in E− (*p*
*<* 0.05). Under the 50 µM CdCl_2_ condition, the proportion of F_NaCl_ and F_HAC_ forms of Cd in E+ plants was significantly (*p*
*<* 0.05) higher than that in E−, and the proportion of F_H2O_ form of Cd was significantly (*p*
*<* 0.05) lower than that in E− (Figure 7). Under the 100 µM CdCl_2_ condition, the proportion of F_Ethanol_ form of Cd in E+ plants was significantly (*p*
*<* 0.05) higher than that in E−, and the proportion of F_H2O_ and F_NaCl_ forms of Cd was significantly (*p*
*<* 0.05) lower than that in E− (Figure 7). In terms of the concentration of different forms of Cd, CdCl_2_ stress had no significant (*p*
*>* 0.05) effect on the F_HCl_ and F_R_ forms of Cd concentration, and there was no significant (*p*
*>* 0.05) difference between E+ and E− (Table 6). F_Ethanol_ and F_NaCl_ forms of Cd concentrations at high concentration (CdCl_2_ at 100 µM) showed a significant (*p*
*<* 0.05) difference between E+ and E− (Table 6). With the increase of CdCl_2_ stress, E+ plants were significantly (*p*
*<* 0.05) lower than E− in the F_H2O_ form of Cd concentration. The F_HAc_ form of Cd concentration was significantly (*p*
*<* 0.05) higher than the control under 100 µM CdCl_2_ stress, and there was no significant (*p*
*>* 0.05) difference between E+ and E− (Table 6).

## 4. Discussion

Previous studies have shown that heavy metal stress can significantly inhibit plant growth while E+ plants are significantly better than E− [34,35,36]. However, in our study, there was no significant difference between E+ and E− plants (Figure 2), probably due to the difference of plant physiological state, different endophytic fungal strains, the influence of external environmental conditions, or the advantages of endophytic fungi to plants being offset by Cd stress. The mechanism behind these findings needs further exploration.

Cd has a negative impact on plant chlorophyll content. Leaf chlorosis is a common poisoning symptom when plants are exposed to heavy metals [45]. An excess amount of Cd hinders the synthesis of chlorophyll by destroying the biosynthetic mechanism and key enzymes of chlorophyll [46]. In the current study, endophyte-infected wild barley significantly improved chlorophyll content at high concentrations of Cd (Figure 2F), showed better chlorophyll functions, and significantly reduced Cd stress. These results were in accordance with the work of Zhang et al. [33] and Zamani et al. [47]. As chlorophyll is the base of plant photosynthesis, any reduction in chlorophyll would significantly affect the photosynthesis of plants. In fact, E+ plants were found to have a higher net photosynthetic rate, transpiration rate, and stomatal conductance (Figure 3), which was consistent with the results of Renet et al. [32].

Endophytes improve the ability of plants to absorb heavy metals from soil and aqueous solution and could make plants accumulate more Cd [31,48]. In our study, the test plants were exposed to Cd solution with increased concentration, resulting in a significant increase in plant parts and total Cd content (Table 3). The Cd content of the E+ plant in the stem was significantly higher than that of the E−, whereas the other parts showed a similar trend but the difference was not significant (Table 3). This might be due to the non-uniform density of endophytic fungal hyphae in the vegetative organs of the plant. The order from high to low absorption was recorded as: stem internode > leaf sheath > leaf, while no hyphae were found in the root [49,50]. Hence, the occurrence of this phenomenon might be related to the distribution characteristics of endophytic fungal hyphae.

The existence of endophytes showed no significant effect on the transport index under the same treatment, which was consistent with the results of Soleimani et al. (2010) [31], and TI_stems-leaves_ were twice as large as TI_roots-stems_ (Table 3). The assumption that the “stem-leaf” transport of Cd promoted by the mycelium distribution in stems would be dense has yet to be confirmed.

The transport of Cd to the shoots was driven by transpiration [51], and the transpiration rate and transport coefficient were affected by CdCl_2_ stress, and both showed that E+ was higher than E− (Figure 3, Table 3). Hence, E+ plants accumulated more Cd compared to E−, which is consistent with the other results of this experiment. In addition, phytochelatin syntheses (PCs) are considered to be the detoxification mechanism of intracellular heavy metals, maintaining the long-distance transport of Cd in plants [52]. The transportation of Cd in plants affects the distribution proportion of different parts. The current study showed that the distribution proportion of Cd in roots was the highest, followed by stems, and lowest in leaves (Figure 4). This is because roots are in direct contact with heavy metals, which are then transported to stems and leaves through various ways, resulting in this decreasing relationship. Of course, the accumulation of heavy metals in roots is considered as a strategy for plants to tolerate heavy metals, which is conducive to the normal physiological and biochemical processes of aboveground plants, such as photosynthesis, respiration, water metabolism, and cell division [53,54]. The proportion of Cd in the roots of E+ plants is less than E− and more than E− in the stems and leaves (Figure 4), indicating that endophytic fungi promote the transportation of Cd to the aboveground parts of plants, so as to reduce the content of heavy metals in aqueous solution and achieve the effect of removing Cd ions.

The roots were in direct contact with CdCl_2_ solution, and thus most seriously affected. The proportion of subcellular distribution in roots was significantly different with the change of CdCl_2_ concentration, but there was no significant difference in stems and leaves (Figure 5). Cell wall fraction and soluble fraction are the main sites for storing Cd. The isolation of cell walls and the compartmentalization of vacuoles are of great significance for plants to resist metal stress [7,9]. The results of this study were consistent with Xin et al. [55] and Zhou et al. [56]. The soluble fraction of the root accounted for 40.47~70.33%, the organelle fraction was 10.43~29.08%, and the cell wall fraction accounted for 19.24~30.45% (Figure 5). As the Cd accumulated in the cell wall reached the threshold under Cd stress, the excess Cd was concentrated in the soluble fraction to reduce the damage of the organelles to ensure normal work. However, Wang et al. [8] showed that the accumulation of Cd in the cell wall of *Bechmeria nivea*. was the highest, followed by the soluble and organelles fraction. He et al. [57] found that Cd accounted for the largest proportion in the cell wall in the subcellular distribution of *Ricinus communis*. Although there are few differences between the reported and the current results, it is certain that whether Cd is concentrated in the cell wall fraction or soluble fraction, its purpose is to protect organelles, thereby protecting the entire plant.

Endophytes showed a significant effect on the distribution proportion of Cd in root organelles. With the increase of CdCl_2_ concentration, E+ plants were significantly higher than E− (Figure 5). Therefore, the root cells of E+ plants showed less obstruction to Cd^2+^ transport to aboveground parts, resulting in the high Cd content in different parts of E+ plants, which is consistent with the results of Cd content in different parts of plants.

The migration ability, toxicity, and biological function of Cd depend on its chemical form in the plants, and there are differences among different chemical forms. Water soluble Cd in the form of inorganic salt (extracted by 80% ethanol) and organic acid (extracted by d-H_2_O) have strong migration ability and the greatest toxicity to plants, followed by pectinate and Cd bound to protein or adsorbed (extracted by 1 M NaCl), while oxalate bound Cd (extracted by 0.6 M HCl) and residual Cd are the least harmful to plants [8,44,58]. The distribution proportion of chemical forms of different Cd in different organs are different. In this study, F_Ethanol_ and F_H2O_ Cd were dominant in leaves, but CdCl_2_ stress had no significant effect on them. The distribution proportion of Cd in F_HAc_, F_HCl_, and F_R_ with low toxicity in stems was significantly higher than that in leaves and roots, and the damage was less, which might be one of the mechanisms of stem resistance against heavy metal stress. The F_Ethanol_, F_H2O_, and F_NaCl_ forms were used to adapt to adverse conditions (Figure 7).

Previous studies have shown that Cd stress can increase Cd in pectin and protein binding forms in stems and leaves, while water-soluble inorganic salts and organic acid Cd are mainly in roots [59,60]. The results of our study were also similar, probably due to the plants’ physiological and biochemical reactions or environmental pollution (soil, solution, external interference, etc.). This approach shows that the isolation of cell wall and vacuole in the CdCl_2_ environment represents the key mechanism for the tolerance and detoxification of wild barley [8,61,62]. The presence of endophytes under Cd stress reduces the proportion of water-soluble Cd in roots (Figure 7), thus reducing plant damage, leading to the accumulation of heavy metals in the roots, stems, and leaves of E+ plants being higher (Table 6), which is consistent with the above research results.

## 5. Conclusions

Plant growth and photosynthesis were inhibited under CdCl_2_ stress. Endophytes increased the chlorophyll content and photosynthetic efficiency of plants and improved the ability of plants to accumulate more Cd in roots, stems, and leaves, indicating that *H. brevisubulatum*–*Epichloë bromicola* symbionts had a strong ability to remove Cd from an aqueous solution. The order of root Cd content in subcellular fractions was: soluble fraction > cell wall fraction > organelle fraction. The isolation of cell wall and vacuole represents a double barrier to protect plant cells. The higher organelle fraction in the roots of E+ plants makes it transport more Cd to the aboveground part, which is of great significance to improve the remediation of heavy metal polluted soil and polluted water. Inorganic salt and organic acid forms accumulated Cd in the leaves and inorganic salt, organic acid, pectin, and protein forms accumulated Cd in roots, while the proportion of phosphate, oxalate, and residual Cd in stems increased to reduce plant damage. The endophytes protect themselves by reducing Cd, which is more toxic to plants.

## Figures and Tables

**Figure 1 jof-08-00366-f001:**
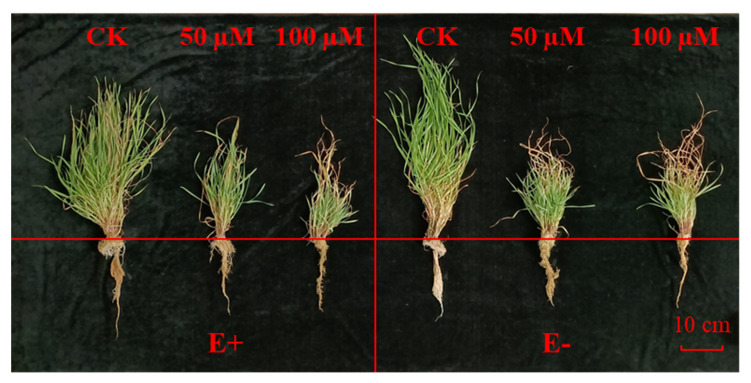
*Hordeum brevisublatum* after 6 weeks of CdCl_2_ stress at different concentrations. E+: endophyte-infected and E−: endophyte-free.

**Figure 2 jof-08-00366-f002:**
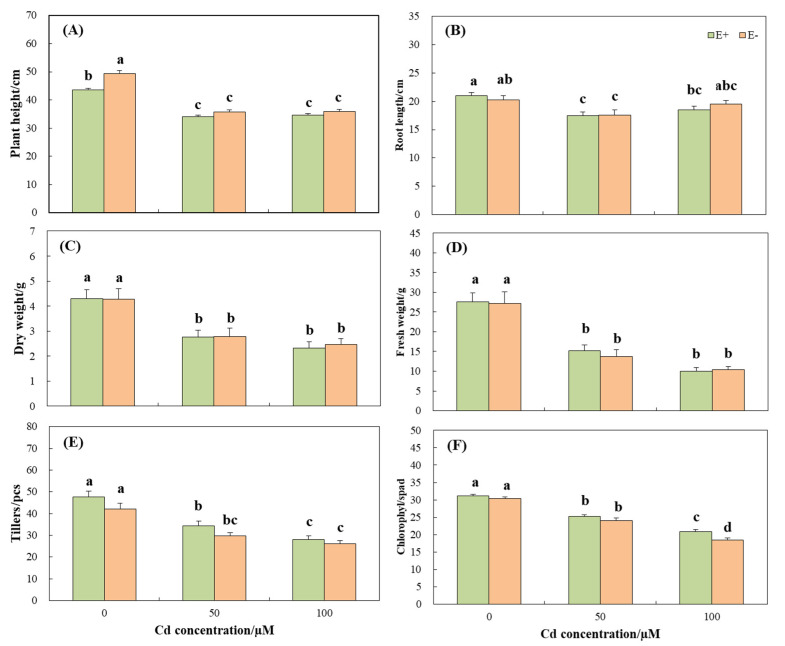
Growth parameters of *Hordeum brevisublatum* under different concentrations of CdCl_2_ stress. (**A**) Plant height, (**B**) Root length, (**C**) Dry weight, (**D**) Fresh weight, (**E**) Tillers numbers, and (**F**) Chlorophyll content. E+: endophyte-infected and E−: endophyte-free. The values presented are mean ± standard error (SE). Lowercase letters (a–d) on top of bars indicate significant differences (*p* < 0.05) of E+ and E− plants under the different concentrations of CdCl_2_ stress.

**Figure 3 jof-08-00366-f003:**
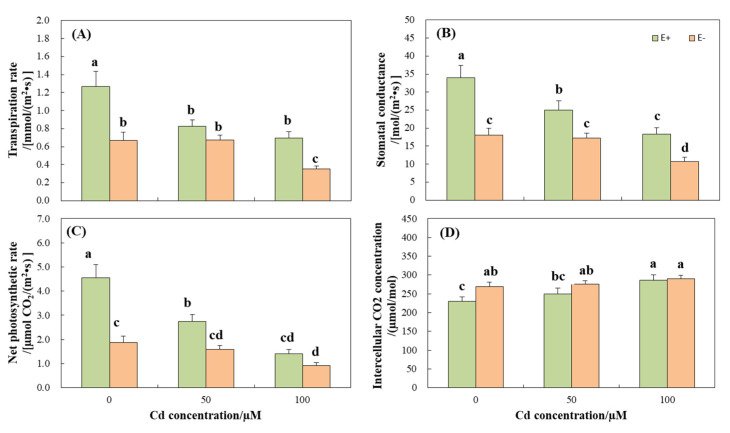
Photosynthetic parameters of *Hordeum brevisublatum* under different concentrations of CdCl_2_ stress. (**A**) Transpiration rate, (**B**) Stomatal conductance, (**C**) Net photosynthetic rate, and (**D**) Concentration of CO_2_ intercellular. E+: endophyte-infected and E−: endophyte-free. The values presented are mean ± standard error (SE). Lowercase letters (a–d) on top of the bars indicate significant differences (*p* < 0.05) of E+ and E− plants under the different concentrations of CdCl_2_ stress.

**Figure 4 jof-08-00366-f004:**
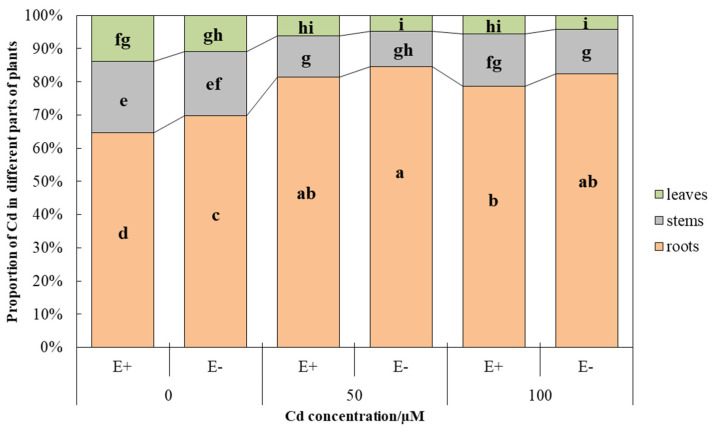
Cd distribution in different parts of *Hordeum brevisublatum* under different CdCl_2_ concentration stress. Lowercase letters (a–i) in the middle of bars indicate significant differences (*p* < 0.05) of E+ and E− plants under the different concentrations of CdCl_2_ stress.

**Figure 5 jof-08-00366-f005:**
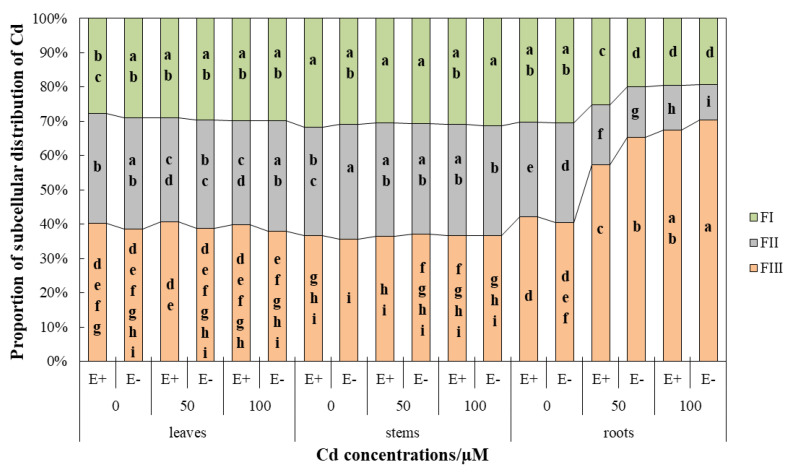
Subcellular proportions of Cd in the leaves, roots, and stems of *Hordeum brevisublatum*. FI: cell wall component, FII: organelle fraction, and FIII: soluble fraction. Lowercase letters (a–i) in the middle of bars indicate significant differences (*p* < 0.05) of E+ and E− plants in different cell components under the different concentrations of CdCl_2_ stress.

**Figure 6 jof-08-00366-f006:**
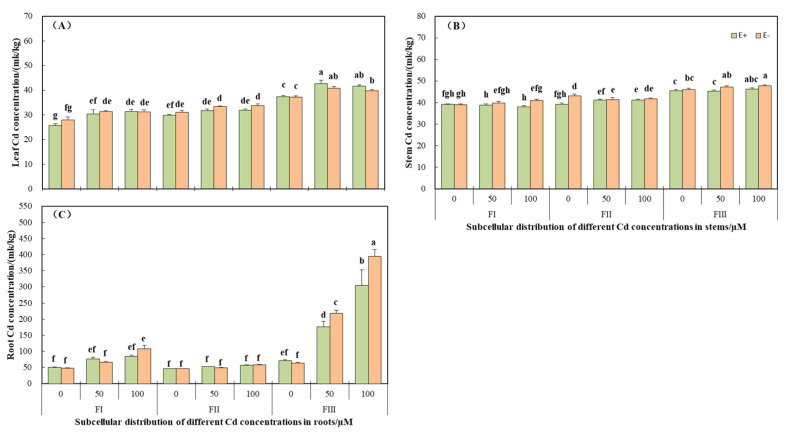
Subcellular distribution of Cd in leaves, stems, and roots of *Hordeum brevisublatum*. (**A**) Leaf, (**B**) Stem, and (**C**) Root. FI: cell wall component, FII: organelle fraction, FIII: soluble fraction, E+: endophyte-infected, and E−: endophyte-free. The values presented are mean ± standard error (SE). Lowercase letters (a–h) on top of the bars indicate significant differences (*p* < 0.05) of E+ and E− plants under the different concentrations of CdCl_2_ stress.

**Figure 7 jof-08-00366-f007:**
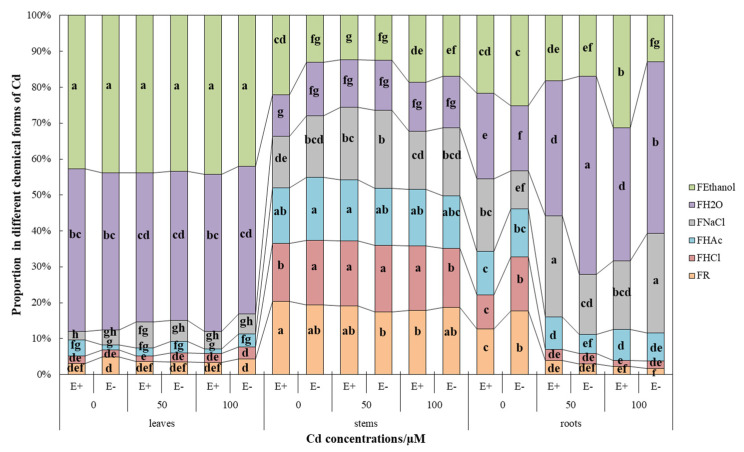
Distribution proportion of different chemical forms of Cd in the leaves, roots, and stems of *Hordeum brevisublatum*. F_Ethanol_: 80% ethanol extraction state; F_H2O_: deionized water extraction state, F_NaCl_: 1M NaCl extraction state, F_HAc_: 2% HAc extraction state, F_HCl_: 0.6 M HCl extraction state and F_R_: residual state. Lowercase letters (a–h) in the middle of bars indicate significant differences (*p* < 0.05) of E+ and E− plants in different chemical forms under the different concentrations of CdCl_2_ stress.

**Table 1 jof-08-00366-t001:** Two-way ANOVA for the effects of endophyte (E) and CdCl_2_ stress(S) on plant height, root length, tiller numbers, Chlorophyll content, dry weight, fresh weight, of *Hordeum brevisublatum*.

Variable	df	Plant Height	Root Length	Tiller Numbers	Chlorophyll Content	Dry Weight	Fresh Weight
E	1	24.383 ***	0.078 ^ns^	5.699 *	9.304 **	0.036 ^ns^	0.109 ^ns^
S	2	162.748 ***	9.444 ***	37.291 ***	190.589 ***	19.152 ***	48.189 ***
E × S	2	6.350 **	0.726 ^ns^	0.398 ^ns^	1.188 ^ns^	0.045 ^ns^	0.125 ^ns^

The numeric data in the Table is F-value; *, **, *** and ns represent significant at *p* ≤ 5%, 1%, 0.1% levels and not significant, respectively.

**Table 2 jof-08-00366-t002:** Two-way ANOVA for the effects of endophyte (E) and CdCl_2_ stress(S) on photosynthetic parameters of *Hordeum brevisublatum*.

Variable	df	Transpiration Rate	Stomatal Conductance	Net Photosynthetic Rate	Intercellular CO_2_ Concentration
E	1	23.429 ***	34.459 ***	36.075 ***	5.391 *
S	2	11.651 ***	14.272 ***	24.628 ***	5.284 ***
E × S	2	2.957 ^ns^	2.440 ^ns^	7.350 ***	1.205 ^ns^

The numeric data in the Table is F-value; *, *** and ns represent significant at *p* ≤ 5%, 1%, 0.1% levels and not significant, respectively.

**Table 3 jof-08-00366-t003:** Cd concentrations in different parts (leaf, stem, root) and the whole of E+ and E− Wild Barley under stress with different CdCl_2_ concentrations, and the transport index from root to stem (TI_roots-stems_) and from stem to leaf (TI_stems-leaves_).

CdCl_2_ Stress/µM	Endophyte	Total/(mg/kg)	Leaf/(mg/kg)	Stem/(mg/kg)	Root/(mg/kg)	TI_roots-stems_/%	TI_stems-leaves_/%
0	E+	142.94 ± 18.86 c	19.48 ± 3.73 c	30.35 ± 3.68 e	93.11 ± 13.75 c	33.79 ± 3.85 a	63.25 ± 8.14 a
	E−	109.98 ± 5.26 c	11.99 ± 0.32 c	20.67 ± 2.14 e	77.32 ± 7.03 c	28.09 ± 5.18 ab	60.35 ± 8.02 a
50	E+	1029.12 ± 24.02 b	63.63 ± 5.68 ab	126.86 ± 9.35 c	838.63 ± 19.39 b	15.08 ± 0.8 6c	52.13 ± 9.00 ab
	E−	906.81 ± 76.55 b	43.62 ± 4.14 b	95.01 ± 3.79 d	768.19 ± 70.67 b	12.55 ± 0.66 c	46.85 ± 3.93 ab
100	E+	1520.43 ± 169.76 a	81.96 ± 15.44 a	230.71 ± 17.29 a	1207.76 ± 178.40 a	20.39 ± 3.47 bc	35.32 ± 5.33 b
	E−	1490.85 ± 114.89 a	61.87 ± 5.04 ab	196.44 ± 16.25 b	1232.55 ± 107.78 a	16.20 ± 1.52 c	32.50 ± 4.66 b

Lowercase letters indicate significant (*p* < 0.05) differences between different CdCl_2_ concentrations for the same index.

**Table 4 jof-08-00366-t004:** Two-way ANOVA for the effects of endophyte (E) and CdCl_2_ stress(S) on Cd concentrations in different parts (leaf, stem, root) and the whole, Root to stem transport index (TI_roots-stems_) and Stem to leaves transport index (TI_stems-leaves_) of *Hordeum brevisublatum*.

Variable	df	Total	Leaf	Stem	Root	TI_roots-stems_	TI_stems-leaves_
E	1	0.70 ^ns^	6.92 *	8.41 **	0.08 ^ns^	2.70 ^ns^	0.52 ^ns^
S	2	118.73 ***	30.12 ***	155.73 ***	80.63 ***	16.57 ***	8.44 ***
E × S	2	0.17 ^ns^	0.48 ^ns^	0.81 ^ns^	0.14 ^ns^	0.13 ^ns^	0.04 ^ns^

The numeric data in the Table is F-value; *, **, *** and ns represent significant at *p* ≤ 5%, 1%, 0.1% levels and not significant, respectively.

**Table 5 jof-08-00366-t005:** Two-way ANOVA for the effects of endophyte (E) and CdCl_2_ stress(S) on subcellular distribution of Cd in leaves, stems, and roots of *Hordeum brevisublatum*.

Variable	df	Leaf	Stem	Root
E	1	0.14 ^ns^	3.17 ^ns^	0.47 *
S	2	3.99 *	0.19 ^ns^	9.19 ***
E × S	2	0.08 ^ns^	0.05 ^ns^	0.33 ^ns^

The numeric data in the Table is F-value; *, *** and ns represent significant at *p* ≤ 5%, 0.1% levels and not significant, respectively.

**Table 6 jof-08-00366-t006:** Concentrations of chemical forms of Cd in leaves, stems and roots of *H. brevisubulatum*.

Chemical Forms	CdCl_2_ Stress/µM	Endophyte	Leaf /(mg/kg)	Stem /(mg/kg)	Root /(mg/kg)
F_Ethanol_	0	E+	23.60 ± 0.37 c	7.79 ± 0.95 ab	10.70 ± 0.59 i
		E−	25.28 ± 0.33 bc	4.36 ± 0.29 mno	9.65 ± 0.56 i
	50	E+	27.69 ± 1.11 a	4.10 ± 0.34 no	30.55 ± 8.53 fg
		E−	26.82 ± 0.36 ab	4.69 ± 0.08 klmno	34.88 ± 0.44 fg
	100	E+	26.08 ± 0.97 ab	6.50 ± 0.47 cdefg	99.17 ± 3.79 c
		E−	26.04 ± 0.43 ab	6.45 ± 0.52 cdefg	54.09 ± 7.08 de
F_H2O_	0	E+	24.93 ± 0.21 bc	3.97 ± 0.22 o	11.92 ± 0.57 i
		E−	25.14 ± 0.41 bc	5.00 ± 0.16 ijklmno	7.00 ± 1.02 i
	50	E+	26.27 ± 0.50 ab	4.41 ± 0.20 lmno	62.19 ± 9.84 d
		E−	25.54 ± 0.27 b	5.24 ± 0.28 hijklmn	114.28 ± 10.60 b
	100	E+	25.83 ± 0.18 ab	4.79 ± 0.26 jklmno	117.91 ± 4.49 b
		E−	25.50 ± 0.15 b	5.43 ± 0.23 ghijklm	200.32 ± 19.72 a
F_NaCl_	0	E+	1.28 ± 0.15 ghi	5.03 ± 0.16 ijklmno	10.05 ± 0.71 i
		E−	2.54 ± 1.89 efghi	5.74 ± 0.44 fghijk	4.07 ± 0.24 i
	50	E+	4.54 ± 0.86 d	6.75 ± 0.89 bcdef	44.93 ± 2.62 ef
		E−	3.60 ± 0.53 de	8.16 ± 0.49 a	34.49 ± 5.42 fg
	100	E+	2.88 ± 0.55 defg	5.63 ± 0.21 fghijk	61.76 ± 9.21 d
		E−	3.50 ± 0.20 def	7.21 ± 0.38 abc	115.38 ± 5.65 b
F_HAc_	0	E+	2.60 ± 1.64 efghi	5.40 ± 0.26 ghijklm	6.19 ± 1.08 i
		E−	0.81 ± 0.20 hi	5.86 ± 0.27 efghijk	5.11 ± 0.22 i
	50	E+	1.42 ± 0.68 ghi	5.68 ± 0.29 fghijk	14.71 ± 1.53 hi
		E−	2.01 ± 0.37 efghi	5.97 ± 0.10 defghij	10.78 ± 2.45 i
	100	E+	0.75 ± 0.38 i	5.50 ± 0.36 ghijklm	27.8 ± 4.32 gh
		E−	2.24 ± 0.43 efghi	5.56 ± 0.10 fghijkl	32.82 ± 6.21 fg
F_HCl_	0	E+	1.16 ± 0.12 ghi	5.65 ± 0.17 fghijk	4.71 ± 0.05 i
		E−	1.13 ± 0.35 ghi	6.00 ± 0.29 defghi	5.75 ± 0.16 i
	50	E+	0.97 ± 0.21 ghi	5.98 ± 0.19 defghij	4.97 ± 0.24 i
		E−	1.56 ± 0.26 fghi	6.94 ± 0.33 bcde	6.03 ± 0.27 i
	100	E+	1.43 ± 0.37 ghi	6.28 ± 0.09 cdefgh	5.51 ± 0.48 i
		E−	2.12 ± 0.11 efghi	6.24 ± 0.17 cdefgh	8.49±0.79 i
F_R_	0	E+	1.72 ± 0.21 efghi	7.09 ± 0.22 abcd	6.30 ± 0.22 i
		E−	2.83 ± 0.15 defgh	6.45 ± 0.23 cdefg	6.73 ± 0.42 i
	50	E+	2.31 ± 0.22 efghi	6.33 ± 0.30 cdefgh	6.11 ± 0.14 i
		E−	2.16 ± 0.18 efghi	6.55 ± 0.23 cdefg	6.04 ± 0.09 i
	100	E+	2.01 ± 0.22 efghi	6.24 ± 0.09 cdefgh	6.97 ± 0.19 i
		E−	2.67 ± 0.20 defghi	7.08 ± 0.36 abcd	6.98 ± 0.17 i

F_Ethanol_: 80% ethanol extraction state; F_H2O_: deionized water extraction state; F_NaCl_: 1M NaCl extraction state; F_HAc_: 2% HAc extraction state; F_HCl_: 0.6 M HCl extraction state; F_R_: residual state; Lowercase letters indicate significance (*p* < 0.05) differences between different CdCl_2_ concentrations for the same index.

**Table 7 jof-08-00366-t007:** Two-way ANOVA for the effects of endophyte (E) and CdCl_2_ stress(S) on chemical forms of Cd in leaves, stems and roots of *Hordeum brevisublatum*.

Variable	df	Leaf	Stem	Root
E	1	0.014 ^ns^	2.934 ^ns^	1.206 ^ns^
S	2	0.093 ^ns^	1.368 ^ns^	22.532 ***
E × S	2	0.016 ^ns^	2.451 ^ns^	0.650 ^ns^

The numeric data in the Table is F-value; *** and ns represent significant at *p* ≤ 0.1% levels and not significant, respectively.

## Data Availability

Not applicable.

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
