# Peer review of "Effect of Fungal Endophyte Epichloë bromicola Infection on Cd Tolerance in Wild Barley (Hordeum brevisubulatum)"

_jof, 2022, doi:10.3390/jof8040366_

Round 1
Reviewer 1 Report
Introduction. Explain and further detail the composition and function of Epichloë endophyte infection in plants, by example: Epichloë endophytes are a group of clavicipitaceous fungi (Clavicipitaceae, Ascomycota) that form symbiotic associations (symbiota) with a broad spectrum of grasses. These biotrophic fungi systemically colonise the intercellular spaces of leaf primordia, leaf sheaths and culms of vegetative tissue. During the reproductive phase of growth, the sexual species form reproductive structures (stromata) around the developing inflorescence that partially or completely sterilize the host. The major benefits for grass endophytes are access to nutrients from the apoplast and a means of dissemination through the seed. Benefits to the host include protection from mammalian and insect herbivory, resistance to nematodes and some fungal pathogens, drought tolerance and greater field persistence. (Scott B. (2001). Epichloë endophytes: fungal symbionts of grasses, Current Opinion in Microbiology, 4(4): 393-398 https://doi.org/10.1016/S1369-5274(00)00224-1)
Material and Methods
Line 120 and others lines: 4°C must be 4 °C “space between number and degrees”
Line 125 and others lines: 3L must be 3 L “space between number and L”
Line 133 and others lines: 20mM must be 20 mM “space between number and mM”
Line 149 and others lines: 0.5g must be 0.5 g and 100ml must be 100 ml. 10ml must be 10 ml
Update reference: Scott B. (2001). Epichloë endophytes: fungal symbionts of grasses, Current Opinion in Microbiology, 4(4): 393-398 https://doi.org/10.1016/S1369-5274(00)00224-1
Author Response
Point 1: Introduction. Explain and further detail the composition and function of Epichloë endophyte infection in plants, by example: Epichloë endophytes are a group of clavicipitaceous fungi (Clavicipitaceae, Ascomycota) that form symbiotic associations (symbiota) with a broad spectrum of grasses. These biotrophic fungi systemically colonise the intercellular spaces of leaf primordia, leaf sheaths and culms of vegetative tissue. During the reproductive phase of growth, the sexual species form reproductive structures (stromata) around the developing inflorescence that partially or completely sterilize the host. The major benefits for grass endophytes are access to nutrients from the apoplast and a means of dissemination through the seed. Benefits to the host include protection from mammalian and insect herbivory, resistance to nematodes and some fungal pathogens, drought tolerance and greater field persistence.
Response 1: Thank you for your valuable comments. I strongly agree with the comments you made. A detailed explanation of the content of endophytic fungi will give the reader a clearer and more comprehensive understanding of the background of this study. Therefore, the introduction part is supplemented and highlighted in yellow (third paragraph, lines 1-7)
Point 2: Material and Methods
Line 120 and others lines: 4°C must be 4 °C “space between number and degrees”
Line 125 and others lines: 3L must be 3 L “space between number and L”
Line 133 and others lines: 20mM must be 20 mM “space between number and mM”
Line 149 and others lines: 0.5g must be 0.5 g and 100ml must be 100 ml. 10ml must be 10 ml
Response 2: Thank you for your valuable comments. I quite agree with you. This detail was ignored by me. I checked the full text and modified it one by one.
Note: the appendix is the revised manuscript

Reviewer 2 Report
Dear Corresponding author
I read your paper and I have some comments that I think may help to improve your paper quality.
You used Epichloë bromicola as fungal symbiont of Hordeum.
You receive the fungal strain from which resource?
You recovered it from a substrate? If yes, you have to have molecular identification and you need to submit its sequences to GenBank and imply the accession number in your paper.
And if you took it from a culture collection you need again imply accession number(s) of sequence(s) in your paper.
Finally you have to produce a phylogenetic tree with the position of your isolate inside of it.
By the way, it is better you insert some plant photos in different experiments such as cd stress tolerance of plants in presence of Epichloe endophyte fungal isolate.

Author Response
Point 1:
You used Epichloë bromicola as fungal symbiont of Hordeum.
You receive the fungal strain from which resource?
You recovered it from a substrate? If yes, you have to have molecular identification and you need to submit its sequences to GenBank and imply the accession number in your paper.
And if you took it from a culture collection you need again imply accession number(s) of sequence(s) in your paper.
Finally you have to produce a phylogenetic tree with the position of your isolate inside of it.
Response 1: Thank you for your valuable comments. I strongly agree with the comments you made. Wild barley seeds were provided by the Institute of Grassland Conservation, College of Grassland Agricultural Science and Technology, Lanzhou University, and the sequence numbers have been indicated in the manuscript. I quoted Chen's article with a detailed phylogenetic tree. The changes are highlighted in green (section "Materials and Methods", first paragraph, lines 2-3).
Point 2: It is better you insert some plant photos in different experiments such as cd stress tolerance of plants in presence of Epichloe endophyte fungal isolate.
Response 2: Thank you for your valuable comments. I quite agree with you. The photos of the plants in the experiment are a more visual representation of the effects of Cd stress on the plants; And it is also a strong evidence of the experiment. Therefore, in the "results" section of the manuscript, I added "Figure 1. Hordeum brevibulatum after 6 weeks of CdCl2 stress at different concentrations."
Point 3: About the "Similarity Report" issue
Response 3: Thank you very much for your work on my manuscript "Similarity Report". I modified the similar parts as much as I could and highlighted them in pink. However, there are some unavoidable duplicates, such as the annotation of pictures or tables, conjunctions, references, etc. , which I had to choose to discard.
Note: the appendix is the revised manuscript

Round 2
Reviewer 2 Report
Dear corresponding author
The paper is now qualified for publication.
Congtatulations.
Reviewer